# The Detection of CMV in Saliva Can Mark a Systemic Infection with CMV in Renal Transplant Recipients

**DOI:** 10.3390/ijms20205230

**Published:** 2019-10-22

**Authors:** Shelley Waters, Silvia Lee, Megan Lloyd, Ashley Irish, Patricia Price

**Affiliations:** 1School of Biomedical Science, Curtin University, Bentley 6102, Australia; shelley.waters@postgrad.curtin.edu.au (S.W.); silvia.lee@curtin.edu.au (S.L.); 2Department of Microbiology and Infectious Diseases, Pathwest Laboratory Medicine, Murdoch 6150, Australia; 3School of Medical and Health Sciences, Edith Cowan University, Joondalup 6027, Australia; megan.lloyd@uwa.edu.au; 4School of Biomedical Sciences, University of Western Australia, Nedlands 6009, Australia; 5Renal Unit, Fiona Stanley Hospital, Murdoch 6150, Australia; Ashley.Irish@health.wa.gov.au; 6School of Medicine and Pharmacology, University of Western Australia, Nedlands 6009, Australia

**Keywords:** cytomegalovirus infection, saliva, renal transplantation

## Abstract

Human cytomegalovirus (CMV) is often transmitted through saliva. The salivary gland is a site of CMV replication and saliva can be used to diagnose congenital CMV infections. CMV replication is monitored in whole blood or plasma in renal transplant recipients (RTR) and associates with clinical disease. However, these assays may not detect replication in the salivary gland and there is little data linking detection in saliva with systemic infection and clinical sequelae. RTR (*n* = 82) were recruited > 2 years after transplantation. An in-house quantitative PCR assay was used to detect CMV UL54 in saliva samples. CMV DNA was sought in plasma using a commercial assay. Vascular health was predicted using flow mediated dilatation (FMD) and plasma biomarkers. CMV-reactive antibodies were quantified by ELISA and circulating CMV-specific T-cells by an interferon-γ ELISpot assay. Vδ2^−^ γδ T-cells were detected using multicolor flow cytometry reflecting population expansion after CMV infection. The presence of CMV DNA in saliva and plasma associated with plasma levels of antibodies reactive with CMV gB and with populations of circulating Vδ2^−^ γδ T -cells (*p* < 0.01). T-cells reactive to CMV immediate early (IE)-1 protein were generally lower in patients with CMV DNA in saliva or plasma, but the level of significance varied (*p* = 0.02–0.16). Additionally, CMV DNA in saliva or plasma associated weakly with impaired FMD (*p* = 0.06–0.09). The data suggest that CMV detected in saliva reflects systemic infections in adult RTR.

## 1. Introduction

Human cytomegalovirus (CMV) is a beta-herpesvirus carried by ~83% of the global adult population [1]. In healthy individuals, most acute infections are asymptomatic but chronic infections have been linked with diseases of aging, notably cardiovascular disease (CVD) [2]. In individuals with acquired immunodeficiencies, such as organ transplant recipients, CMV can cause acute end-organ disease and/or contribute to risk of graft rejection, CVD and secondary bacterial and fungal infections [3]. Symptomatic CMV infections occurs in 20–60% of transplant recipients depending on donor and recipient CMV status, type of organ transplanted, degree of immunosuppression and use of anti-viral prophylaxis [4].

The burden of CMV in an individual can be measured directly by the detection of CMV DNA or indirectly via specific immune responses (antibodies in plasma or CMV-specific T-cells). IgG antibodies reactive with CMV assessed in plasma or serum reflect a lifetime history of infection. High titers of CMV-reactive IgG antibodies have been associated with all-cause mortality, development of CVD and reduced responses to influenza vaccination in elderly populations [5]. Similarly, high CMV antibody levels in individuals living with human immunodeficiency virus (HIV) are associated with cerebrovascular disease and CVD [6]. However, the interpretation of antibodies in this context is confounded because antibody levels increase as HIV patients achieve an immunological response to antiretroviral therapy [7]. Once HIV patients are stable on therapy, higher T-cell responses (particularly CD8^+^ T-cell responses to immediate early (IE)-1 protein) suggest a high burden of CMV [8]. However, this response may represent a lifetime of persistent infections that are predominantly latent, as in older CMV seropositive adults, up to 23% of the entire CD8^+^ T-cell compartment can be CMV-specific [9].

It may also be possible to assess the CMV burden based on novel populations of natural killer (NK) and T-cells that have been linked with CMV seropositivity and constitute an “immunological footprint” [10]. This includes a subset of gamma delta (γδ) T-cells. γδ T-cells constitute < 5% of circulating T-cells in healthy adults [11]. Most are Vγ9^+^ and Vδ2^+^, while Vδ2^−^ (mainly Vδ1^+^ and Vδ3^+^) cells predominate in mucosal epithelia [12]. Populations of Vδ2^−^ γδ T-cells are expanded in renal transplant recipients (RTR) and healthy adults who are CMV-seropositive [13].

RTR who are seronegative for CMV and receive an organ from a seropositive individual have a high risk of clinical sequelae [4] and are routinely managed with anti-viral medication prophylaxis to prevent clinical disease [14]. Following transplantation, CMV DNA is routinely monitored in plasma or whole blood; whole blood may offer greater sensitivity [15,16,17]. However congenital CMV is usually monitored using urine or saliva according to clinical practice guidelines [18]. CMV replicates in acinar cells of the salivary gland and saliva is a common route of transmission [19]. Saliva is easily collected as a non-invasive sample, but the value of detecting CMV in the saliva of adults is not well understood.

The present study utilizes clinical and immunological measures of the systemic footprint of CMV to assess the value of detecting CMV DNA in saliva samples compared with a commercial assay based on CMV DNA in plasma.

## 2. Results

### 2.1. Frequency of CMV DNA Detection in Saliva and Plasma

RTR were screened for the presence of CMV DNA in saliva samples using an in-house qPCR assay and in plasma using a commercial assay (Abbot Molecular). CMV DNA was detected in 11 (13%) saliva samples and 16 (21%) plasma samples. Nine individuals had detectable CMV DNA in both saliva and plasma, so detection of CMV DNA in saliva was more common in RTR who had CMV DNA in their plasma (*p* < 0.0001).

### 2.2. CMV DNA Detected in Saliva is Associated with Immunological Responses to CMV

All comparisons are shown in Appendix A and informative comparisons are presented in Figure 1. The presence of CMV DNA in saliva or plasma from RTR associated with plasma levels of CMV antibodies detected with gB antigen (Figure 1A, *p* = 0.009 and Figure 1B, *p* = 0.006) and populations of Vδ2^−^ γδ T-cells (Figure 1C, *p* = 0.01 and Figure 1D, *p* = 0.005). Presence of CMV DNA in saliva also associated with increased T-cell responses to the VLE peptide (Figure 1G, *p* = 0.02) which is a component of the IE-1 antigen. T-cell responses to IE-1 peptide pool followed a similar pattern (Figure 1E, *p* = 0.14). The presence of CMV DNA in plasma associated with increased T-cell responses to IE-1 peptides (Figure 1F, *p* = 0.04) and generally higher VLE-specific T-cell responses (Figure 1H, *p* = 0.16). T-cell responses to the NLV peptide were higher in individuals carrying CMV DNA in saliva (Figure 1K, *p* = 0.03) and followed a similar trend in patients with CMV DNA in plasma (Figure 1L, *p* = 0.54). However, one patient with CMV DNA in saliva and high NLV-specific T-cell responses had no CMV DNA in plasma detected with the Abbot Molecular qPCR assay, so this did not approach significance. There were no associations with antibodies targeting CMV lysate or IE-1, T-cell responses to CMV lysate or pp65 pooled peptides, or inflammatory biomarkers (Appendix A).

### 2.3. CMV DNA Displayed Weak Positive Associations with Cardiovascular Risk

The presence of CMV DNA in saliva or plasma associated weakly with inferior flow mediated dilatation (FMD) (Figure 2A, *p* = 0.087 and Figure 2B, *p* = 0.062). There were no associations with carotid intima media thickness (cIMT) (*p* > 0.52; Appendix A) but biomarkers associated with CVD showed some consistent trends. The presence of CMV DNA in plasma associated with plasma levels of VCAM-1 (Figure 2D, *p* = 0.03), with a similar trend to levels of ICAM-1 (Appendix A). Accordingly, high VCAM-1 correlated weakly with reduced FMD (*p* = 0.04, r = −0.24), whilst there was no correlation between ICAM-1 and FMD. The pattern was similar when CMV DNA was assessed in saliva, but the trends were not significant (Figure 1C *p* = 0.27 and Figure 1D *p* = 0.20, respectively). Additionally, levels of *p*-selectin in plasma were lower in RTR with CMV DNA in saliva or plasma (*p* = 0.01).

## 3. Discussion

Our study evaluated the utility of detecting CMV DNA in saliva samples from RTR. Saliva was pursued as a non-invasive alternative to blood because it may capture bursts of CMV replication in acinar cells [20]. CMV replication can be missed when assessed in plasma from HIV patients with CMV disease [21]. We provide evidence that detection of CMV DNA in saliva is reflective of systemic immune responses to the virus, marked by antibody and T-cell responses. The associations mirror those seen when CMV DNA was detected in plasma with a clinical assay used to monitor CMV after transplantation.

The presence of CMV DNA at either site associated with levels of gB antibody, but not with antibody reactive with CMV lysate or IE-1. There is considerable evidence that gB is a major target for the immune system [22,23]. Moreover, gB is a vaccine candidate and a recent longitudinal study of this cohort suggests that antibodies against gB may be protective against deterioration of cardiovascular health measured by FMD (Affandi et al., submitted for publication). One seronegative individual tested positive for CMV DNA in saliva but was not assessed by the Abbot assay. The patient did not have detectable IgM reactive with CMV gB in plasma or saliva (data not shown). However, the transplanted kidney was from a seropositive donor and it is notable that seroconversion did not occur. Overall, 26 plasma and seven saliva samples were assessed for IgM reactive with CMV gB. Only one sample had detectable IgM and it was present in both plasma and saliva (data not shown). This suggests that new infections are rare—a contention supported by assessments of γδ T-cells.

The presence of CMV DNA in saliva or plasma was also associated with similarly elevated proportions of Vδ2^−^ γδ T-cells. These comprise cells expressing Vδ1, Vδ3 and Vδ5. In our cohort, Vδ2^−^ γδ T-cells represent around 0.5% of the T-cell population in CMV-seronegative RTR and 5–10% in seropositive RTR [13]. The likelihood that detection of CMV DNA several years after transplantation is due to reactivation rather than a primary infection may explain the clear association between increased populations of Vδ2^−^ γδ T-cells and CMV DNA positivity. Similar findings were noted by Pitard et al. who linked maximal changes in T-cell populations with reactivation rather than primary infection. They compared donor negative/recipient positive (D^−^/R^+^) renal transplants resulting in a CMV reactivation event with D^+^/R^−^ transplants where there was a primary CMV infection [24].

Detection of CMV DNA in plasma associated significantly with T-cell (CD4^+^ and CD8^+^) responses to the IE-1 peptide pool and weakly with CD8^+^ T-cell responses to the IE-1 peptide, VLE. Detection of CMV DNA in saliva associated significantly with T-cell responses to VLE and weakly with responses to the IE-1 peptide pool. In RTR, strong T-cell responses to IE-1 may decrease graft rejection and improve graft function [25]. IE-1 is the first protein expressed during reactivation, so IE-1-reactive T-cells may control bursts of replication quickly. CMV DNA in saliva also associated with increased CD8^+^ T-cell responses to the pp65 peptide, NLV, but not with T-cell responses to pp65 peptide pools (Figure 1I–L). Accordingly, Leng et al. associated NLV T-cell responses with the presence of CMV DNA from peripheral blood mononuclear cells (PBMC) in adults over 70 years of age [26].

In this cohort, levels of CMV antibody were an independent marker of reduced FMD [27] and CMV DNA in both saliva and plasma aligned weakly with impaired FMD. FMD is an assessment of endothelial function based on the ability of the artery to respond to shear stress [28]. Associations between CMV and cardiovascular disease have been validated in a meta-analysis [2]. Here, CMV DNA in plasma also associated with levels of VCAM-1, and levels of ICAM-1 followed a similar trend. VCAM-1 and ICAM-1 are cell adhesion molecules that bind to integrins to aid in the transcellular migration of leukocytes. Both are expressed in atherosclerotic lesions [29,30].

*p*-selectin is a cell adhesion molecule expressed on platelets and endothelial cells. *p*-selectin is implicated in the formation of atherosclerotic lesions in mice [31] and myocardial infarction in humans [32]. Paradoxically, CMV DNA in plasma or saliva associated with decreased levels of *p*-selectin in plasma. Moreover, elevated plasma *p*-selectin associated with increased FMD in healthy adults recruited in parallel with the RTR described herein (Affandi et al., submitted for publication). Studies of cell-bound and soluble *p*-selectin are required to unravel the pathways invoked.

The current study assessed the systemic effect of detecting CMV DNA in saliva in RTR who were >2 years post-transplant and clinically stable at the time of recruitment. Futures studies are needed to address the clinical utility of detecting CMV DNA in saliva in individuals at risk of active disease.

## 4. Materials and Methods

### 4.1. Study Cohort

Eighty-two RTR were recruited from renal clinics at Royal Perth Hospital (Western Australia). Inclusion criteria were clinical stability >2 years after transplant, no CMV disease or reactivation within 6 months of sample collection and no current anti-viral treatment. RTR infected with hepatitis B or C were excluded. Ethics approval was obtained from Royal Perth Hospital Human Research Ethics Committee (approval number: EC 2012/155) and endorsed by Curtin University Human Research Ethics Committee (approval number: HR16/2015). Participants provided written informed consent.

### 4.2. Detection of CMV DNA in Plasma and Saliva

Saliva (approximately 5 mL) was collected after a water mouth wash by asking the participant to spit into a 50 mL centrifuge tube. Samples were centrifuged (1000× *g*, 10 min). DNA was extracted from the pellet using FavorPrep Blood Genomic DNA Extraction Mini Kits (Favorgen, Ping-Tung, Taiwan) and stored at −80 °C. CMV was detected using an in-house qPCR assay with primers targeting the UL54 gene (CMV DNA polymerase) [7]. Quantitation was achieved using a standard curve created using DNA extracted from a lysate of CMV (AD169)-infected HFF which was serially diluted 10-fold. Samples were considered positive if a steady amplification curve was initiated before 38 cycles (a cut-off based on the lowest point on the standard curve). Positive results were normalized against the gene encoding beta-2-microgobulin and values were reported in arbitrary units (range: 44–721). CMV DNA was also detected in EDTA plasma using the Abbot Molecular assay (Abbot Laboratories, Chicago, IL, USA) in the Department of Microbiology, Royal Perth Hospital (Western Australia). Five seronegative plasma samples were not assessed using the Abbot Molecular assay and were excluded from analyses. The Abbot assay reported samples as either “Not detected”, “< 20 copies/mL” or as a viral load. Samples with viral loads or reported as <20 copies/mL were analyzed as CMV DNA positive. The two assays do not provide viral loads on the same scale; analyses compare samples grouped as CMV DNA positive or negative.

### 4.3. Immunological Assessments of CMV Burden

Peripheral blood mononuclear cells (PBMC) were isolated by Ficoll density centrifugation. Plasma was stored in −80 °C and PBMC in liquid nitrogen. Plasma CMV IgG titers were assessed using in-house ELISAs based on a lysate of fibroblasts infected with CMV AD169, recombinant CMV gB (Chiron Diagnostics, Medfield, MA, USA) or IE-1 protein (Miltenyi Biotech, Cologne, Germany). Results are presented as arbitrary units (AU)/mL based on a standard plasma pool [23], allowing comparisons between people but not between antigens. Plasma levels of vascular biomarkers (*p*-selectin, ICAM-1 and VCAM-1) and inflammatory biomarkers (sTNFR1, sCD14 and CRP) were quantified using commercial ELISA antibody pairs (R&D Systems, Minneapolis, MN, USA).

PBMC were used to assess T-cell responses to CMV lysate and peptide pools derived from pp65 and IE-1 (JPT Peptide Technologies; Berlin, Germany) via ELISpot assay. These antigens and peptide pools are known to stimulate CD4^+^ and CD8^+^ T-cell responses. CD8^+^ T-cell responses to NLV and VLE peptides (derived from pp65 and IE-1, respectively) were assessed in samples from individuals who carried human leukocyte antigen (HLA)-A2. PBMCs were also used to enumerate Vδ2^−^ γδ T-cells using multicolor flow cytometry, as the population is expanded in CMV-seropositive RTR [13].

### 4.4. Assessment of Vascular Pathology

Ultrasonography was used to assess carotid intima media thickness (cIMT) and flow mediated dilatation (FMD) of the brachial artery after 10 min of rest [33]. cIMT is a measurement of the thickness of the inner layer (intima) of the carotid artery and is a marker of subclinical atherosclerosis. FMD assesses the ability of the larger conduit artery to respond to shear stress via endothelial-dependent and -independent mechanisms.

### 4.5. Statistical Analyses

Mann–Whitney non-parametric statistics and Fisher′s exact tests utilized GraphPad Prism version 8 for Windows (Graphpad Software, La Jolla CA, USA). Comparisons achieving *p* < 0.05 were considered significant while comparisons achieving 0.05 < *p* < 0.1 are noted as a trend.

## 5. Conclusions

This study addressed the utility of detecting CMV DNA in saliva in RTR compared with assays using plasma in clinical settings. We show that CMV DNA detected in saliva reflects systemic infection as assessed by antibody and T-cell responses and note a trend of impaired cardiovascular health assessed by FMD when CMV DNA was present at either site.

## Figures and Tables

**Figure 1 ijms-20-05230-f001:**
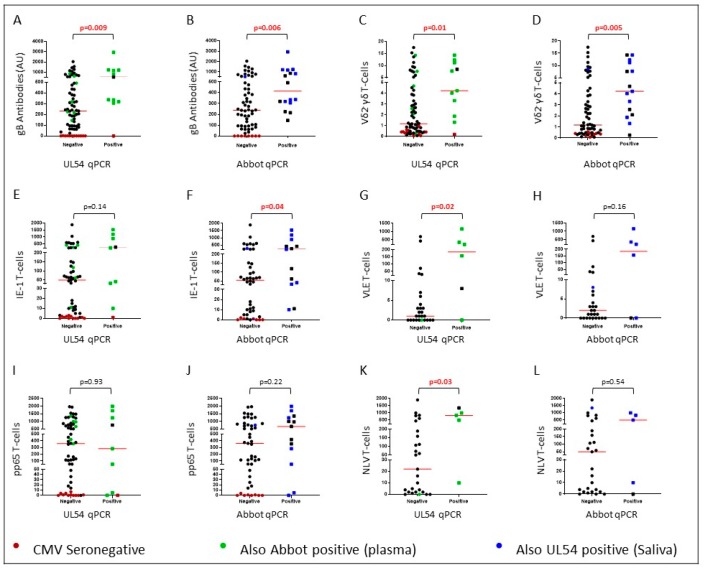
Human cytomegalovirus (CMV) DNA was detected using an in-house qPCR targeting UL54 in saliva or a commercial assay (Abbot Molecular) in plasma. Plots (**A**) and (**B**) compare levels of gB reactive antibodies in plasma. Plots (**C**) and (**D**) compare populations of Vδ2^−^ γδ T-cells as a percentage of CD3^+^ cells. Plots (**E**) and (**F**) compare T-cell responses to the immediate early (IE)-1 antigen. Plots (**G**) and (**H**) compare T-cell responses to the VLE peptide. Plots (**I**) and (**J**) compare T-cell responses to the pp65 antigen. Plots (**K**) and (**L**) compare T-cell responses to the NLV peptide reported as interferon-γ spot forming units per 200,000 cells. Points colored red represent CMV seronegative individuals.

**Figure 2 ijms-20-05230-f002:**
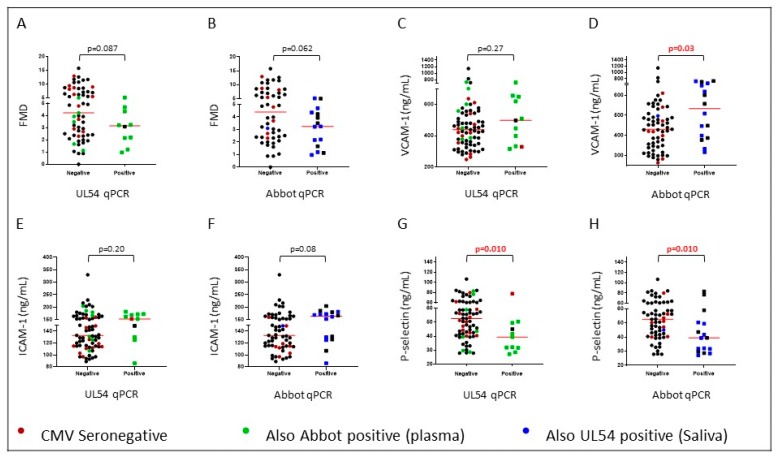
DNA was detected using an in-house qPCR targeting UL54 in saliva or a commercial assay (Abbot Molecular) in plasma. Plots (**A**) and (**B**) compare flow mediated dilatation (FMD). Plots (**C**) and (**D**) compare levels of VCAM-1 in plasma. Plots (**E**) and (**F**) compare levels of ICAM-1 in plasma. Plots (**G**) and (**H**) compare levels of *p*-selectin in plasma. Points colored red represent CMV seronegative individuals.

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
