# Peer review of "The Detection of CMV in Saliva Can Mark a Systemic Infection with CMV in Renal Transplant Recipients"

_ijms, 2019, doi:10.3390/ijms20205230_

Round 1

Reviewer 1 Report

The work by Kaushik et al. employs different approaches to quantify the presence of HCMV in in renal transplant recipients (RTR) and correlates HCMV infection with immunological responses and cardiovascular risk. Given the clinical significance of HCMV infection and the need of new biomarkers that can help to predict the outcome of infection, these data provide relevant insights for the management of RTRs.

I think that the analyses are very detailed and in my opinion, the paper can be published after a minor revision. Below my specific comments:

HCMV viral load values in the different samples should be reported (not just presence/absence) and verify if there is a statistically significant correlation between viral load values in different samples, immunological responses and cardiovascular risk. The clinical information related to the patients should be better specify, possibly in a dedicated table, including age, sex, origin, kind of transplant (D-/R+; D+/R+), concomitant pathologies etc… Please include the complete name of the ethics committee that approved the study in your revised manuscript and the number of the approval. Report also the CMV IgM levels in these patients. It is very important to understand if patients enrolled in the study have a past infection or a reactivation. It would be useful to report the serostatus of the patients enrolled in this study also for other blood-transmissible infections (such as HIV, HBV, HBC, syphilis), and that they were exempt from any pathology or treatment which could interfere with inflammatory and metabolic parameters (e.g. history of cancer, autoimmune diseases, active or recent systemic infections), to confirm that the observed results are specifically related to HCMV. Throughout the paper, as well as in the abstract, actual values are not presented when describing findings. More details should be included…for example, what does “marginally associated” means (line 27)?

Author Response

The work by Kaushik (Waters) et al. employs different approaches to quantify the presence of HCMV in in renal transplant recipients (RTR) and correlates HCMV infection with immunological responses and cardiovascular risk. Given the clinical significance of HCMV infection and the need of new biomarkers that can help to predict the outcome of infection, these data provide relevant insights for the management of RTRs.  I think that the analyses are very detailed and in my opinion, the paper can be published after a minor revision. Below my specific comments:

HCMV viral load values in the different samples should be reported (not just presence/absence) and verify if there is a statistically significant correlation between viral load values in different samples, immunological responses and cardiovascular risk.

Intuitively this seems like a better strategy but there are several problems

Too many zeros. As discussed on page 5, most samples had no viral DNA detectable by either assay. Moreover several plasma samples assessed by the Abbott assay were only reported to us as <20 copies /ml. We suspect that viral loads change rapidly over time, so the reading on a given day may have little meaning. The two assays do not provide viral loads on the same scale so the sample types can’t be compared.

These considerations also invalidate the use of correlation coefficients to compare viral loads with immunological responses and cardiovascular risk.

The text on page 5 has been modified to clarify the assays used.

The clinical information related to the patients should be better specify, possibly in a dedicated table, including age, sex, origin, kind of transplant (D-/R+; D+/R+), concomitant pathologies etc…

Age and sex already in supplementary table 1 and  time since transplantation has been added.

Although we were underpowered to assess clinical factors, we noted that patients had a range of underlying pathologies that included membranoproliferative glomerulonephritis, IgA nephropathy, polycystic kidney disease and focal segmental glomerulosclerosis. None associated with significantly detection of CMV DNA. Additionally, the kind of transplant did not associate with CMV DNA positivity.

Please include the complete name of the ethics committee that approved the study in your revised manuscript and the number of the approval. Report also the CMV IgM levels in these patients.

Full name of ethics committees and approval number has been added on page 5.

A subset of 26 plasmas and 7 saliva supernatants were also assessed for IgM specific for gB (data not shown). Only one sample had detectable IgM and it was present in both plasma and saliva. We have noted this in the discussion.

It is very important to understand if patients enrolled in the study have a past infection or a reactivation. It would be useful to report the serostatus of the patients enrolled in this study also for other blood-transmissible infections (such as HIV, HBV, HBC, syphilis), and that they were exempt from any pathology or treatment which could interfere with inflammatory and metabolic parameters (e.g. history of cancer, autoimmune diseases, active or recent systemic infections), to confirm that the observed results are specifically related to HCMV.

The text already specifies RTR with hepatitis B or C were excluded. No individuals in the study had a history of syphilis or HIV past or current. This has been confirmed by the renal physician associated with the study (AI).

Throughout the paper, as well as in the abstract, actual values are not presented when describing findings. More details should be included…for example, what does “marginally associated” means (line 27)?

The data are provided in the Tables. Repetition in the text would be very cumbersome, but we would accept an editorial decision that this is warranted. P values are provided in the text. We now better define the term “marginally associated” in the M&M (page 6) as 0.05<p<0.1.

Reviewer 2 Report

General Comments.

This paper seeks to determine whether detection of CMV DNA in saliva might provide a useful diagnostic measure of virus load and disease.  It is worthwhile to assess how it stands up against the conventional plasma qPCR as it offers a considerable advantage to the patient.

The data suffers from a paucity of PCR+ve samples from either qPCR test.  There is reassurance that most of the samples that were PCR+ve in the saliva were also positive in the blood, but there are nevertheless queries as to why a saliva PCR+ve sample might be concomitantly negative for the blood.  As suggested below, it would be useful to show the actual DNA loads in these samples rather than just “positive” or “negative”. 

Given the paucity of positive samples, correlation with various biomarkers needs to be interpreted with caution.  In general, the results text reflects a cautious appraisal of the data.  However, the discussion should be toned done somewhat; I don’t believe it is useful to summarise associations as “marginal” or “weak” as it diminishes the rigour of the work.  Most peer reviewers accept that a low number of clinical samples often precludes being able to make clear associations; it is best to leave it at that.

Specific points

The title of the paper is a little misleading; it suggests that CMV in the saliva correlates with high systemic infection, but the paper offers no data concerning the systemic load sustained by the RTR cohort, nor if the saliva CMV+ individuals had high systemic loads.

Given that much of the focus concentrates on associations of biomarkers with responses/disease, it is unfortunate that DNA levels in saliva or plasma were presented as simply “positive” or “negative”.  It would be useful for the reader to gain some idea of the virus loads and their range in the cohort. 

Did the authors perform any preliminary studies to assess the sensitivity/reproducibility of their in-house qPCR assay? 

Samples tested by either the in-house or commercial kit came from the same RTR cohort, and most of these (of the order of 80%) were negative.  Of the positive samples, it is stated (line 73) that 9 out of the 11 individuals that were CMV+ positive for saliva also had detectable CMV DNA in plasma.  Why is there a correlative disparity between qPCR assay type and some biological parameters (VLE and NLV T-cells, VCAM).  Is this due to the few samples that were qPCR positive by one assay only?    It would be helpful to identify the individuals in the plots of Figs 1 and 2 that were positive by one or both of the methods using colour-coded dots, particularly in cases where there is a disparity in the correlative outcome between the in-house and commercial PCR tests AND where only a proportion of samples were tested.  It would also be helpful for the y-axes to be equivalent between qPCR groups. 

There is one CMV-seronegative subject that was positive by the UL54 qPCR.    Is this potentially a primary infection (since serum measured IgG and not IgM responses) or a false positive? 

Line 16: I don’t recall any evidence that the salivary gland is a site of latency.

Line 20:  Should “sought” replace “quantified” in this sentence?

Line 24: Replace CVM with CMV

Line 26:  Consider replacing the word “proportions” here; ?

The number of samples tested for the inflammatory or vascular markers in supplementary Table 1 is absent.

Line 131 “This may explain the clear association…”  It is not clear what the association with CMV DNA is here or what “effect” is stronger.

Line 204: “Moreover the presence of CMV DNA at either site was associated with measures of cardiovascular health assessed by FMD”.  Is this accurate?  This does not reflect the data presented in Fig 2A and B.

Author Response

General Comments.

This paper seeks to determine whether detection of CMV DNA in saliva might provide a useful diagnostic measure of virus load and disease.  It is worthwhile to assess how it stands up against the conventional plasma qPCR as it offers a considerable advantage to the patient.

The data suffers from a paucity of PCR+ve samples from either qPCR test.  There is reassurance that most of the samples that were PCR+ve in the saliva were also positive in the blood, but there are nevertheless queries as to why a saliva PCR+ve sample might be concomitantly negative for the blood.  As suggested below, it would be useful to show the actual DNA loads in these samples rather than just “positive” or “negative”. 

See response to Reviewer 1

Given the paucity of positive samples, correlation with various biomarkers needs to be interpreted with caution.  In general, the results text reflects a cautious appraisal of the data.  However, the discussion should be toned done somewhat; I don’t believe it is useful to summarise associations as “marginal” or “weak” as it diminishes the rigour of the work.  Most peer reviewers accept that a low number of clinical samples often precludes being able to make clear associations; it is best to leave it at that.

We have added to the text “Comparisons achieving 0.05<p<0.1 are noted as marginal if they support a trend.” We are reluctatnt to conclude that comparisons yielding p=0.05 mark a difference whereas those with p=0.06 mark identity.

Specific points

The title of the paper is a little misleading; it suggests that CMV in the saliva correlates with high systemic infection, but the paper offers no data concerning the systemic load sustained by the RTR cohort, nor if the saliva CMV+ individuals had high systemic loads.

The title is changed to: The detection of CMV in saliva can mark a systemic infection with CMV in renal transplant recipients

Given that much of the focus concentrates on associations of biomarkers with responses/disease, it is unfortunate that DNA levels in saliva or plasma were presented as simply “positive” or “negative”.  It would be useful for the reader to gain some idea of the virus loads and their range in the cohort. 

See response to Reviewer 1

Did the authors perform any preliminary studies to assess the sensitivity/reproducibility of their in-house qPCR assay? 

Sensitivity of the assay was assessed by extracting DNA from a lysate of CMV (AD169) -infected fibroblasts (HFF). The DNA was serially diluted 10-fold, and detection of amplicons after 38 cycles was recorded. The serial dilutions were used to create a standard curve that was run alongside each batch of samples. All samples, controls and standards were assessed in duplicate and the same positive control (a CMV DNA positive saliva sample from a different cohort) was included and consistently amplified on the same cycle. Page 5 has been edited to explain this with greater clarity.

Since this work was completed, all CMV DNA positive samples have been amplified with a nested PCR targeting UL55 (glycoprotein B) for our sequencing studies.

Samples tested by either the in-house or commercial kit came from the same RTR cohort, and most of these (of the order of 80%) were negative.  Of the positive samples, it is stated (line 73) that 9 out of the 11 individuals that were CMV+ positive for saliva also had detectable CMV DNA in plasma.  Why is there a correlative disparity between qPCR assay type and some biological parameters (VLE and NLV T-cells, VCAM).  Is this due to the few samples that were qPCR positive by one assay only?    It would be helpful to identify the individuals in the plots of Figs 1 and 2 that were positive by one or both of the methods using colour-coded dots, particularly in cases where there is a disparity in the correlative outcome between the in-house and commercial PCR tests AND where only a proportion of samples were tested. 

Colour coded dots have been added to show which samples are positive in each assay.

It would also be helpful for the y-axes to be equivalent between qPCR groups. 

All y-axses have been altered so they are broken at the same points and each section shares the same percentage of the y-axis.

There is one CMV-seronegative subject that was positive by the UL54 qPCR.    Is this potentially a primary infection (since serum measured IgG and not IgM responses) or a false positive? 

Text has been added to discussion (page 4).

This individual received a kidney from a seropositive donor, so it is notable that he did not seroconvert. However, we did not detect any IgM or IgG in plasma or saliva (data not shown). It is also not a false positive result as we have confirmed the presence of CMV using PCRs targeting 3 other genes (UL55, UL18 and UL40) which were sequenced using Sanger methodologies. We have also recently acquired next generation sequence of his CMV using Ampliseq methods. Analses are underway.

Line 16: I don’t recall any evidence that the salivary gland is a site of latency.

Published papers that document the salivary gland as a site of latency use mouse models (so they are looking at MCMV). It is less clear in humans so text has been edited.

Line 20:  Should “sought” replace “quantified” in this sentence?

Line 20 already says “sought”. No change is made as we don't report quantitation.

Line 24: Replace CVM with CMV                   Done

Line 26:  Consider replacing the word “proportions” here; ?

The word “proportions” has been replaced with “populations”

The number of samples tested for the inflammatory or vascular markers in supplementary Table 1 is absent.

A footnote has been added to clarify that numbers are provided when some samples could not be assessed.

Line 131 “This may explain the clear association…”  It is not clear what the association with CMV DNA is here or what “effect” is stronger. Text has been modified to clarify this.

Line 204: “Moreover the presence of CMV DNA at either site was associated with measures of cardiovascular health assessed by FMD”.  Is this accurate?  This does not reflect the data presented in Fig 2A and B. Text has been modified to clarify this.